# Inhibiting Metastasis and Improving Chemosensitivity via Chitosan-Coated Selenium Nanoparticles for Brain Cancer Therapy

**DOI:** 10.3390/nano12152606

**Published:** 2022-07-29

**Authors:** Paweena Dana, Nuttaporn Pimpha, Angkana Chaipuang, Nutthanit Thumrongsiri, Prattana Tanyapanyachon, Anukul Taweechaipaisankul, Walailuk Chonniyom, Natsorn Watcharadulyarat, Sith Sathornsumetee, Nattika Saengkrit

**Affiliations:** 1National Nanotechnology Center (NANOTEC), National Science and Technology Development Agency (NSTDA), Pathum Thani 12120, Thailand; paweena.dan@nanotec.or.th (P.D.); nuttaporn@nanotec.or.th (N.P.); angkana.cha@ncr.nstda.or.th (A.C.); nutthanit.thu@nanotec.or.th (N.T.); prattana.tan@nanotec.or.th (P.T.); anukul.taw@ncr.nstda.or.th (A.T.); walailuk.cho@ncr.nstda.or.th (W.C.); natsorn.wat@ncr.nstda.or.th (N.W.); 2Research Network NANOTEC-Mahidol University in Theranostic Nanomedicine, Faculty of Medicine Siriraj Hospital, Mahidol University, 2 Wanglang Road, Bangkoknoi, Bangkok 10700, Thailand; sith.sat@mahidol.edu; 3Department of Medicine (Neurology), Faculty of Medicine Siriraj Hospital, Mahidol University, 2 Wanglang Road, Bangkoknoi, Bangkok 10700, Thailand

**Keywords:** selenium nanoparticle, chitosan, glioma, metastasis, drug sensitivity, blood–brain barrier

## Abstract

Selenium nanoparticles (SeNPs) were synthesized to overcome the limitations of selenium, such as its narrow safe range and low water solubility. SeNPs reduce the toxicity and improve the bioavailability of selenium. Chitosan-coated SeNPs (Cs-SeNPs) were developed to further stabilize SeNPs and to test their effects against glioma cells. The effects of Cs-SeNPs on cell growth were evaluated in monolayer and 3D-tumor spheroid culture. Cell migration and cell invasion were determined using a trans-well assay. The effect of Cs-SeNPs on chemotherapeutic drug 5-fluorouracil (5-FU) sensitivity of glioma cells was determined in tumor spheroids. An in vitro blood–brain barrier (BBB) model was established to test the permeability of Cs-SeNPs. SeNPs and Cs-SeNPs can reduce the cell viability of glioma cells in a dose-dependent manner. Compared with SeNPs, Cs-SeNPs more strongly inhibited 3D-tumor spheroid growth. Cs-SeNPs exhibited stronger effects in inhibiting cell migration and cell invasion than SeNPs. Improved 5-FU sensitivity was observed in Cs-SeNP-treated cells. Cellular uptake in glioma cells indicated a higher uptake rate of coumarin-6-labeled Cs-SeNPs than SeNPs. The capability of coumarin-6 associated Cs-SeNPs to pass through the BBB was confirmed. Taken together, Cs-SeNPs provide exceptional performance and are a potential alternative therapeutic strategy for future glioma treatment.

## 1. Introduction

Glioblastoma (GBM) is the most common and highly aggressive primary brain malignancy in adult humans. Poor quality of life and poor prognosis of GBM patients have been documented [1]. The major problems of GBM treatment are drug resistance and restricted permeation of drugs across the blood–brain barrier (BBB) [2,3]. Currently, selective permeability of the BBB leads to ineffective treatment of GBM. Conventional drug delivery into the targeted brain site is challenging because the BBB restricts the entry of small molecules according to the needs of the central nervous system. According to the literature, only 1% of drugs can pass across the BBB to reach the tumor site, resulting in a poor patient outcome [4,5,6]. Therefore, novel therapeutic strategies to overcome these main obstacles are urgently required.

Selenium (Se) is an essential element and is the most required micronutrient in human health for normal growth and development of biological activity [7,8,9]. Selenium from several natural sources exists in two different forms: inorganic Se (sodium selenite or selenate) and organic Se (mainly SeMet). Selenium can be incorporated into proteins, known as selenoproteins, which significantly contribute to the bioactivity of human and animal health [10,11]. The functions of selenoproteins depend on the presence of selenocysteine at the active site [12]. In the human body, Se exhibits effects that are antioxidant, anti-inflammatory [13,14], antimutagenic [15], anticarcinogenic [16,17], antiviral [9,18], antibacterial [19], and antifungal [20]. Selenium has been studied for the treatment of inflammatory diseases such as rheumatoid arthritis and asthma [21], and for cancer treatment as a chemotherapeutic or radiotherapeutic adjuvant. Compared with normal cells, malignant cells exhibit more oxidative stress; therefore, cancer cells are more susceptible to the pro-oxidant effects of selenium than normal cells [22,23,24].

Traditional uses of Se are risky because of toxicity and the difficulty distinguishing between Se deficiency, appropriate intake, and excess intake [25]. The use of Se-containing compounds as potential chemo-therapeutic agents is limited because high levels of Se supplementation can induce toxicity [23]. However, using the proper amount of Se pro-vides an outstanding benefit for treatment. Since the narrow therapeutic index of Se is a concern, nanotechnology plays a crucial role in developing selenium nanoparticles (SeNPs) to overcome this obstacle by reducing toxicity and improving biocompatibility [23,26]. SeNPs have been applied in several biomedical applications as a drug delivery system, especially as an anti-cancer agent for cancer treatment [25,27,28,29]. The efficiency of SeNPs in inhibiting tumor cell growth in vitro and *in vivo* was demonstrated in cancers such as liver, breast, prostate, lung, and brain cancer [30,31,32,33,34,35]. Compared with the same high dose of soluble Se, which is commonly toxic, SeNPs were found to be effective and well tolerated *in vivo* [36,37]. SeNPs comprise an inorganic therapeutic core of SeO that can be stabilized or functionalized in the loaded active drug or specific compounds [25,38]. For coating or functionalizing SeNP, several approaches have been studied to improve stability and target the therapeutic effect [39]. Numerous compounds have been used to modify the surface of SeNP, such as folic acid, hyaluronic acid, chitosan, polysaccharides, amino acids, peptides, and proteins [40,41,42,43,44].

This study aims to investigate the inhibitory effects of chitosan-coated SeNPs (Cs-SeNPs) on the aggressiveness of GBM cells in terms of cell proliferation, migration, and invasion. In addition, this study treats GBM cells with a combination of Cs-SeNPs and the chemotherapeutic drug 5-fluorouracil (5-FU) to observe cell growth in a 3D-tumor spheroid model, and it offers a novel, alternative therapeutic strategy for the clinical treatment of GBM. The results suggest that Cs-SeNPs effectively inhibit GBM cell proliferation, cell migration, and invasion, and that applying Cs-SeNPs increases the sensitivity of GBM cells to 5-FU.

## 2. Materials and Methods

### 2.1. Materials

Dulbecco’s Modified Eagle Medium (DMEM) (cat. #12800017), Penicillin–Streptomycin–Glutamine (100X) (cat. #10378016), trypsin-EDTA (cat. #25200056), and fetal bovine serum (FBS) (cat. #10270106) were purchased from GIBCO Invitrogen (Grand Island, NY, USA). Matrigel Growth Factor Reduced (cat. #354230) was purchased from Corning (Corning, NY, USA). MTT 3-(4,5-dimethylthiazol-2-yl)-2,5-diphenyltetrazolium bromide (cat. #475989) was purchased from EDM Millipore (Burlington, MA, USA). Dimethyl sulfoxide (DMSO) (cat. #102952) was purchased from Merck Millipore (Sigma Aldrich; Merck Millipore, Darmstadt, Germany). Chitosan, molecular weight of 200,000 Daltons (cat. #9012-76-4), was purchased from Seafresh Chitosan (Lab) Company Limited (Chumphon, Thailand).

### 2.2. Methods

#### 2.2.1. Cell Culture

U87 cells, a glioblastoma cell line, were purchased from the American Type Culture Collection (ATCC number HTB-14, Manassas, VA, USA). Cells were cultured in DMEM complete medium supplemented with 10% FBS, 0.1 mM non-essential amino acids (100 µg/mL l-glutamine), and 100 μg/mL streptomycin and 100 U/mL penicillin. Cells were incubated at 37 °C in a humidified atmosphere containing 5% CO_2_.

#### 2.2.2. Formulations of SeNPs and Cs-SeNPs

For the synthesis of SeNP, 1 mL of 0.1 M sodium selenite solution was prepared. Then, 0.1 M L-ascorbic acid solution was added to the solution. The mixture was stirred for an additional 3 h to obtain the orange–red color liquid phase. Samples were dialyzed overnight against ultrapure water in a dialysis bag (MWCO: 12,000–14,000, T series; Cellusep) at 4 °C to eliminate interference from L-ascorbic acid and its oxidation products.

Cs-SeNPs were synthesized using a similar approach to SeNPs. Ten milliliters of 0.10–0.20% *w*/*v* chitosan was dissolved in 85 mL of distilled water under magnetic stirring at 700 rpm for 30 min. Next, 1 mL of 0.1 M sodium selenite solution was slowly added to the mixture. After 30 min, 4 mL of 0.1 M l-ascorbic acid solution was added dropwise into the mixture. The mixture was then stirred for an additional 3 h to obtain the orange–red color liquid phase. Samples were dialyzed overnight against ultrapure water in a dialysis bag (MWCO: 12,000–14,000, T series; Cellusep) at 4 °C to eliminate interference from L-ascorbic acid and its oxidation products. The concentration of Se was determined by inductively coupled plasma mass spectrometry (ICP-MS; Model 7900, Agilent Technologies, Santa Clara, CA, USA). Dynamic light scattering (DLS, Ultra Pro, Malvern Panalytical Ltd., Malvern, United Kingdom) was used to monitor the change in particle size and zeta potential values during storage. Nanoparticles were diluted with DI water, and 1 mL of each sample was measured in a disposable capillary cell (DTS1070). The size and zeta potentials were also investigated in the water solution within 15 days.

#### 2.2.3. Formulations of Coumarin-6-Labeled SeNPs and Cs-SeNPs

The synthetic procedure is briefly described as follows: Chitosan (1.80 mL, 0.10–0.20% *w*/*v*) was mixed with 15.12 mL of DI water under the rotator mixer for 30 min. Next, 0.18 mL of 0.1 M sodium selenite solution was slowly added to the mixture, and then 0.18 mL of 20 µM coumarin-6 in ethanol was added. After 30 min, 0.72 mL of 0.1 M l-ascorbic acid solution was added dropwise into the mixture. Next, the mixture was rotated overnight to obtain the orange–red color liquid phase. Samples were placed in a dialysis bag (MWCO: 12,000–14,000, T series; Cellu-sep, San Antonio, TX, USA) and dialyzed overnight against ultrapure water at 4 °C to eliminate interference from L-ascorbic acid and its oxidation products. The absolute concentration of Se was determined by inductively coupled plasma mass spectrometry (ICP-MS; Model 7900, Agilent Technologies, Santa Clara, CA, USA). The coumarin-6-labeled SeNPs were synthesized using a similar method except the addition of chitosan.

#### 2.2.4. Physicochemical Characterizations of SeNPs and Cs-SeNPs

The morphology of the SeNPs and Cs-SeNPs was investigated using transmission electron microscopy (TEM). Briefly, samples were prepared by placing 5 µL of the sample onto a 200-mesh carbon copper grid (cat. #EMS200-Cu, Electron Microscopy Sciences, Hatfield, PA, USA). After 10 min, the droplet was removed from the edge of the grid using filter paper and then air dried. The samples were observed using TEM (model JEM 2010; JEOL Inc., Peabody, MA, USA) with an accelerating voltage of 120 kV and magnifications of 25,000× *g*, 60,000× *g*, and 100,000× *g*.

The hydrodynamic diameter, polydispersity index (PDI), and zeta potential of SeNPs and Cs-SeNPs were determined using dynamic light scattering (DLS) (Zetasizer; nanoseries, Malvern; Worcestershire, UK). Then, 20 µL of the sample was diluted in 1 mL of filtered distilled water before measuring to eliminate the viscosity effects. The hydrodynamic diameter, PDI, and zeta potential were obtained from the average of three measurements at 25 °C.

#### 2.2.5. MTT Assay for Cytotoxicity Evaluation of Cs-SeNPs and SeNPs

To evaluate the cytotoxicity of SeNPs and Cs-SeNPs, 100 µL of the U87 cell suspension (8 × 10^3^ cells/well) in DMEM complete medium was seeded into a 96-well plate (cat. #3590, Corning Inc., Corning, NY, USA) and incubated overnight. Cells were treated with either free sodium selenite (Na_2_SeO_3_-), SeNPs, or Cs-SeNPs at various concentrations of selenium (0–25 µg/mL) for 24 h. After the treatments, 10 µL of MTT solution (5 mg/mL) was added. The plates were incubated for 4 h before the medium was removed. Next, 100 µL of DMSO was added to dissolve the formazan crystals. Absorption values at 570 nm were determined by using a microplate reader (SpectraMAX, Molecular Devices, Poway, CA, USA).

#### 2.2.6. Effect of Cs-SeNPs on 3D-Tumor Spheroid Cell Viability

The effect of SeNPs and Cs-SeNPs on 3D spheroid cultures was tested. U87 cells (1000 cells/well) in 100 µL of DMEM complete medium were seeded into 96-well round-bottom ultra-low attachment plates (cat. #7007, Corning Inc., Corning, NY, USA) and incubated for 4 days to introduce spheroids. Afterward, 100 µL of each sample (Na_2_SeO_3_-, SeNPs, and Cs-SeNPs) was added to the tumor spheroids and further incubated for 2 days. Bright-field images of the treated U87 tumor spheroids were taken with an Olympus IX50 inverted microscope (Olympus, Tokyo, Japan).

To investigate the effect of Cs-SeNPs on chemotherapeutic drug sensitivity, a combination of SeNPs or Cs-SeNPs and the chemotherapeutic drug 5-fluorouracil (5-FU) was administered to the tumor spheroid. The U87 spheroids were concomitantly treated with SeNPs or Cs-SeNPs at a concentration of 3.125 µg/mL and with 5-FU at a concentration of 5 µM. After 48 h of incubation, the morphology of the U87 tumor spheroid was observed, and the volume of the spheroid was quantitated using ImageJ software (NIH, Bethesda, MD, USA). The experiments were performed in triplicate.

#### 2.2.7. Quantitative Real-Time PCR

Total RNA was extracted using TRIzol reagent (cat. #15596026, Thermo Fisher Scientific, Waltham, MA, USA), and cDNA was prepared using iScript Reverse Transcription Kits (Bio-Rad Laboratories, Hercules, CA, USA). Quantitative RT-PCR analysis was performed using the Luna Universal qPCR (cat. #M3003L, New England Biolab Inc., Ipswich, MA, USA). The mRNA expression levels of MRCP-1 and BRCP were normalized with Ct of β2Mg and calculated as ΔCt  =  Ct _target_  −  Ct _GAPDH_, and 2^−ΔΔCt^ was used to calculate the fold change. The oligonucleotide primers of MRCP-1 and BRCP used were previously reported [45].

#### 2.2.8. Cell Migration and Invasion

Cell migration and cell invasion of U87 cells were analyzed using a Boyden chamber assay with an uncoated 8.0 μm pore insert (cat. #3422, Corning Inc., Corning, NY, USA) and Matrigel-coated insert (0.4 mg/mL) for migration assay and invasion assay, respectively. The cell suspension (3 × 10^4^ cells) with or without SeNPs or Cs-SeNPs in a serum-free medium was seeded into the chamber and allowed to migrate or invade for 24 h. Cells in the chamber were removed by scraping with a cotton bud, and cells at the lower surface of the insert were fixed and stained with sulforhodamine B (cat. #S1402-1G, Sigma-Aldrich, Darmstadt, Germany). Migrated or invaded cells were counted under the microscope (5 microscopic fields/well).

#### 2.2.9. Gelatin Zymography Assay

The conditioned medium was collected from cells cultured in serum-free DMEM for 24 h. Debris cells were removed by centrifugation at 2000 × *g* at 4 °C for 5 min. The supernatant was collected afterward. The activities of matrix metalloproteinases 2 and 9 (MMP-2 and MMP-9) were determined using a gelatin zymography assay, as described previously [46]. Gels were stained with 0.5% Coomassie brilliant blue G-250 for 30 min and rehydrated in 2% acetic acid overnight at room temperature. The stained bands were scanned using ImageQuant LAS 600 (GE Healthcare, Little Chalfont, UK) and analyzed by ImageQuant TL (GE Healthcare, Little Chalfont, UK).

#### 2.2.10. Cellular Uptake of SeNPs and Cs-SeNPs

To investigate the cellular uptake of the Cs-SeNPs in U87 cells, the coumarin-6-labeled SeNPs and Cs-SeNPs were applied. U87 cells (4 × 10^4^ cells/well) were seeded onto a cover slip in a 24-well plate and cultured overnight. Cells were treated with either coumarin-6-labeled SeNPs or coumarin-6-labeled Cs-SeNPs at a concentration of 6.25 µg/mL, which corresponds with the concentration that is lower than IC_20_ (cell viability > 80%). Cells were then incubated at 37 °C under an atmosphere of 5% CO_2_. After 0.5, 1, and 2 h of incubation time, the cells were washed with phosphate buffer saline and fixed with 4% paraformaldehyde for 20 min at room temperature. The fixed cells were triple washed with PBS and then stained with Hoechst 33,342 dye (0.5 μg/mL) (Invitrogen, Paisley, UK), followed by further incubation for 30 min at room temperature. The stained cells were triple washed with PBS and observed using confocal laser scanning microscopy (CLSM) (FLUOVIEW FV10i, Olympus, Tokyo, Japan) with a 60× objective lens. The fluorescent intensity was calculated relative to the control group using ImageJ software (v1.53s) (NIH, Bethesda, MD, USA).

#### 2.2.11. The *in Vitro* Blood–Brain Barrier Model

Mouse brain microvascular endothelial cells (bEnd.3) were purchased from ATCC. To mimic the BBB, the *in vitro* BBB was established as previously described [47]. Briefly, bEnd.3 cells were seeded to the luminal compartment of the inserts at a density of 8 × 10^4^ cells/cm^2^ of Transwell membrane (0.33 cm^2^ for a 24-well polycarbonate plate, cat. #3413, Corning Inc., Corning, NY, USA). The bEnd.3 cells were cultured in DMEM with 10% FBS for 7 days. Experiments were performed on day 7. Trans-endothelial electrical resistance (TEER) was measured using a Millicell-ERS-2 (Millipore, Billerica, MA, USA). The wells with a TEER value greater than 50 Ω·cm^2^ [48,49] were selected for further analysis of their permeability of coumarin-6-labeled SeNPs or Cs-SeNPs. The coumarin-6-labeled SeNPs or Cs-SeNPs were then added into the luminal compartment of the insert. At the indicated time point (0, 0.5, 1, 4, 6, and 24 h), 200 µL of media in the lower chamber was collected and an equal volume of fresh media was immediately added to the lower chamber. The signal of coumarin-6 in the collected media was determined using a fluorescent spectrometer (SpectraMAX, Molecular Devices, Poway, CA, USA).

#### 2.2.12. Statistical Analysis

The data were expressed as the mean ± standard deviation. The significant differences observed between the experimental groups were determined using Student’s *t*-test, and *p* < 0.05 was considered a statistically significant result. All statistical analyses were performed using SPSS 17.0 (SPSS Inc., Chicago, IL, USA).

## 3. Results

### 3.1. Morphology and Physicochemical Characteristics of SeNPs and Cs-SeNPs

The morphology of SeNPs and Cs-SeNPs prepared by chemical reduction were observed under TEM. The nanoparticles were monitored with magnifications of 25,000× *g* for SeNPs, and 60,000× *g* and 100,000× *g* for Cs-SeNPs, as shown in Figure 1. The results exhibited spherical shapes with sizes under 500 and 100 nm for SeNPs and Cs-SeNPs, respectively. The mean particle diameters and zeta potentials of SeNPs and Cs-SeNPs evaluated by DLS. The average size diameters of SeNPs were 417.60 ± 46.17 nm, which is significantly larger than all conditions of Cs-SeNPs. The diameters of Cs-SeNPs with chitosan contents of 0.1, 0.15, and 0.2% were 88.66 ± 0.65, 88.90 ± 0.57, and 91.45 ± 2.57 nm, respectively. Thus, the sizes of each concentration of chitosan in Cs-SeNPs were not significantly different. For the result of the PDI, the coating of chitosan caused the PDI to decrease from 0.48 ± 0.08 into the range of 0.20 ± 0.01 to 0.23 ± 0.02, which indicates that chitosan promotes particle stabilization. The zeta potentials of Cs-SeNPs were 43.27 ± 3.26, 41.50 ± 2.65, and 47.97 ± 0.80 mV when the chitosan contents were 0.1, 0.15, and 0.2%, respectively, whereas the zeta potential of SeNPs was −0.9 ± 0.66 mV. This result obviously indicates that Cs-SeNPs have a significantly higher positive charge than SeNPs; CS-SeNPs provided strong positive charges to the nanoparticles due to the positive charge of chitosan [50]. Correspondingly, the increasing positive charge of Cs-SeNPs showed a dose-dependent effect. The characteristics of SeNPs and Cs-SeNPs from this work were comparable with previous work that reported the negative charge of SeNPs (−14.8 ± −3.6 mV) and positive charge of Cs-SeNPs (21.0 ± 0.2 mV). The size of SeNPs was also found to be greater than that of Cs-SeNPs [51]. The zeta potential of Cs-SeNPs displays a positive charge because the amino groups on the chitosan surface generate positive zeta potentials. Although the SeNPs showed a negative charge on their outer surfaces, the Cs-SeNPs exhibited a positive charge that can induce higher anticancer activity and intrinsic antimicrobial activity to inhibit the growth of bacteria and fungi [52]. As a result, their smaller size and stronger positive charge may enable Cs-SeNPs to contribute to cancer therapy via higher cellular internalization compared with SeNPs.

The hydrodynamic diameter and zeta potential of SeNPs and Cs-SeNPs were monitored to analyze particle stability within 15 days. The results indicate that coating SeNPs with chitosan enhanced the stability of the nanoparticles regarding the change in size and zeta potential (Figure 2A). The gross appearance of SeNPs and Cs-SeNPs on day 14 were not changed from day 0 (Figure 2B). The charge of SeNPs size diameter fluctuated and the zeta potential was changed in the unstable range. Both parameters were rather stable in the Cs-SeNPs, especially for the zeta potential value. Within 15 days, the zeta potential values of all Cs-SeNPs were higher than 30 mV, indicating the high stability of nanoparticles. Moreover, the increased chitosan likely induces higher stability according to the consistency of size and the increase in zeta potentials.

### 3.2. Evaluation of Antitumor Activity and Enhancing Chemotherapeutic Drug Sensitivity of Cs-SeNPs and SeNPs in Glioma Cells

The antitumor activity of free sodium selenite (Na_2_SeO_3_-), Cs-SeNPs (Cs content of 0.1, 0.15, and 0.2%), and SeNPs was evaluated in the U87 cell line and in normal fibroblasts. The results suggest that the cell viability of U87 was reduced in a dose-dependent manner (Figure 3). Sodium selenite showed an inhibitory effect on U87 but also presented a severe cytotoxic effect on normal fibroblast cells (Figure 3A,B). Using nanoparticle-based selenium, both SeNPs and Cs-SeNPs, is a promising approach to compromise between the bioactivities of toxicity and therapeutic effects. The viability of U87 cells treated with SeNPs was decreased in a dose-dependent manner. At an equal concentration, SeNPs also showed a negative effect on normal fibroblast cells, which is in contrast to previous studies that demonstrated the potency of SeNPs inducing the cytotoxicity against hepatocarcinoma, colorectal cancer, and cervical cancer cell lines, but not against normal cells [53,54,55]. This result implies that different types of cells respond to drugs with different therapeutic doses. In our study, we found that the concentration of SeNPs required to inhibit tumor growth of glioma is too high and harmful for normal fibroblast cells. Therefore, Cs-SeNPs exhibited potential as an alternative therapeutic agent without being harmful to normal cells. Among the different percentages of chitosan, 0.2% CS-SeNPs provided a large difference in toxicity between normal and cancer cells. At the highest concentration (25 ug/mL), 0.2% CS-SeNPs showed the lowest cell viability compared to the other %CS-SeNP. Meanwhile, the lowest effect was found in normal fibroblasts (Figure 3B). Therefore, 0.2% Cs-SeNPs were selected for further experiments to investigate the effect on phenotypic changes in U87 cells. This phenomenon corresponded with previous research which documented that selenium exhibited a narrow therapeutic window between the beneficial and the toxic effect [25,56,57].

### 3.3. Cs-SeNPs Increased Sensitivity of U87 Cells to 5-Fluorouracil

Currently, 5-FU is a first line chemotherapeutic drug used for treating glioma and other cancers. However, drug resistance, toxicity, and adverse effects limit the clinical application of 5-FU and become the major problems resulting in cancer progression. To emphasize the effectiveness of 0.2% Cs-SeNPs, we established 3D-tumor spheroids of U87 to examine antitumor activity and to mimic the conditions of a solid tumor [58]. The spheroids were tested with and without the chemotherapeutic drug 5-FU. Comparing to control spheroid volume, 5 and 10 µM of 5-FU had no effect on reducing tumor spheroid growth of U87 cells (data not shown). Therefore, the combination treatment of Cs-SeNPs with 5-FU in tumor spheroid growth of U87 cells was investigated. Figure 4A shows the results of treatment without 5-FU (upper panel), specifically that the size of the tumor spheroid treated with 0.2% Cs-SeNPs was remarkably smaller than the tumor spheroid treated with SeNPs, especially at the concentrations of 12.5 and 25 µg/mL. 

With 5-FU at a concentration of 5 µM (lower panel), 0.2% Cs-SeNPs at a concentration of 6.25 µg/mL provided a stronger effect on inhibiting tumor spheroid growth than SeNPs at equal concentration. The tumor spheroid volumes were calculated and shown in Figure 4B. Without 5-FU, tumor spheroid volumes of the SeNPs-treated group at the concentrations of 12.5 and 25 µg/mL were 5602266.24 ± 1987537 and 2733933.47 ± 802893.10, respectively. Meanwhile, the tumor spheroid volumes of the Cs-SeNPs-treated group at the concentrations of 12.5 and 25 µg/mL were 3985632.01 ± 269291.16 and 1410681.20 ± 389595.95, respectively. In combination with 5 µM of 5-FU, the tumor spheroid volumes of the SeNPs-treated group at the concentrations of 12.5 and 25 µg/mL were 4051909.11 ± 190726.3 and 3383991 ± 567601.6, respectively. Meanwhile, the tumor spheroid volumes of the Cs-SeNPs-treated group at the concentrations of 12.5 and 25 µg/mL were 2149895 ± 317665.5 and 880786.7 ± 310985.9, respectively. The results demonstrate that coating the SeNPs with chitosan might be an effective alternative therapeutic strategy for treating brain cancer. Our study indicated that SeNPs can enhance chemosensitivity of drugs during therapy, in agreement with previous studies [59,60], and our results also demonstrated that chitosan-coated SeNPs could enhance the sensitivity of the chemotherapeutic drug 5-FU against glioma cells. The presence of chitosan on the particle surface probably promotes cellular internalization of nanoparticles via endocytosis [61].

The tumor can become tolerant to the chemotherapeutic treatment, resulting in drug resistance and eventual tumor recurrence. Several mechanisms in cancer are involved in 5-FU resistance, such as overexpression of drug transporters [62]. In this study we aimed to investigate the role of SeNPs and 0.2% Cs-SeNPs on the expression levels of genes involved in drug resistance in glioma, including MRP1 (ABCC1) and BRCP (ABCG2), which can predict clinical progression. MRP1 and BRCP proteins are involved in plasma membrane efflux pumps that reduce the efflux of intracellular drugs [63,64]. In samples from glioma patients, the p-glycoprotein and MRP1 were expressed at approximately 35% and 50%, respectively [63]. To investigate the mechanism underlying how 0.2% Cs-SeNPs increased sensitivity of 5-FU against U87 cells, we performed the expressions of drug transporters and analyzed them using real-time polymerase chain reaction (RT-PCR). The results revealed that the expression of MRP1 in 0.2% Cs-SeNPs-treated cells was significantly reduced in a dose-dependent manner, while SeNPs-treated cells were unchanged (Figure 4C). This result implies that the chitosan coating on SeNPs increased the capability of SeNPs in treatment. SeNPs and 0.2% Cs-SeNPs, however, cannot alter the gene expression of BRCP, since the expression level of BRCP was comparable between groups treated with SeNPs and 0.2% Cs-SeNPs (Figure 4C). In conclusion, the results of RT-PCR indicate that 0.2% Cs-SeNPs can enhance sensitivity of 5-FU against glioma cells by downregulating expression of MRP1. The results also indicate the benefit of 0.2% Cs-SeNPs as an alternative treatment to enhance drug sensitivity to 5-FU.

### 3.4. Cs-SeNPs Inhibited Cell Migration and Cell Invasion of Glioma Cells

We performed a transwell Boyden chamber assay to evaluate the effect of SeNPs and 0.2% Cs-SeNPs against tumor aggressiveness via inhibition of cell migration and cell invasion of U87 cells. The migrated cells and invaded cells in the lower chamber were stained by sulforhodamine B and observed under a microscope (Figure 5A,B). The result was quantitated and revealed that SeNPs and 0.2% Cs-SeNPs showed an inhibitory effect against cell migration of glioma cells in a dose-dependent manner (Figure 5C). Comparing to control cells, the cell migration analysis obtained from SeNPs at concentrations of 3.125 and 6.25 µg/mL were 80.3 ± 8.9% and 71.3 ± 6.8%, respectively, whereas those of 0.2% Cs-SeNPs were 52.0 ± 0.3% and 49.2 ± 0.8%, respectively (Figure 5C). As a result, the effect of 0.2% Cs-SeNPs on reducing cell migration was significantly higher than that of SeNPs. For the invasion assay, the data revealed that treating cells with concentrations of 3.125 and 6.25 µg/mL of SeNPs resulted in cell invasion of 89.7 ± 7.8% and 90.2 ± 6.5%, respectively, relative to control cells. Those treated with 0.2% Cs-SeNPs at concentrations of 3.125 and 6.25 µg/mL decreased cell invasion significantly to 80.2 ± 0.4% and 67.1 ± 6.2% (Figure 5D). This result indicates that only 0.2% Cs-SeNPs could inhibit the cell invasion ability of glioma cells. Although previous studies indicated that SeNPs can inhibit cell migration and cell invasion of cancer cells [65,66], our study demonstrates that 0.2% Cs-SeNPs have a stronger effect on cell migration and cell invasion than SeNPs.

### 3.5. Cs-SeNPs Inhibited Cell Migration and Cell Invasion of Glioma Cells by Inhibiting MMP-2/9 Activities

To identify the underlying mechanism of 0.2% Cs-SeNPs for inhibiting cell migration and cell invasion, we determined MMP-2 and MMP-9 using a gelatin zymography assay. The results revealed that the activities of MMP-2 and MMP-9 detected from 0.2% Cs-SeNPs-treated cells were dramatically reduced compared with those in the control and SeNPs-treated cells (Figure 6A–C). The results of MMP-2/9 activities corresponded with the effect of Cs-SeNPs on cell migration and invasion activities in U87. Our study agrees well with the reports related to the activity of selenite and SeNPs on decreasing pro-MMP-2/9 in HT1080 (fibrosarcoma) cells and the lung cancer cell line [67].

### 3.6. Chitosan Coating Enhanced Cellular Uptake of Cs-SeNPs in U87 Cells

To emphasize the significance of 0.2% Cs-SeNPs on cancer cell phenotypic changes, U87 cells were incubated with coumarin-6-labeled Cs-SeNPs or coumarin-6-labeled SeNPs for 0.5, 1, and 2 h. The internalization and distribution of nanoparticles in U87 cells were analyzed using CLSM. The confocal images revealed that signals of coumarin-6-labeled Cs-SeNPs were detected at 0.5 and 1 h post-incubation, while SeNPs were hardly observed until 2 h after the signal of coumarin-6-labeled SeNPs was detected (Figure 7A). Quantitative analysis showed that the fluorescent intensities of coumarin-6-labeled SeNPs were significantly lower than those of Cs-SeNPs (Figure 7B). The results indicate that U87 cells allowed the entry of Cs-SeNPs with a higher affinity more than SeNPs at every time point by increasing 9.2-, 2.9-, and 2.4-fold at the time points of 0.5, 1, and 2 h, respectively. The cellular uptake of chitosan-based nanoparticles in U87 cells within 1 h had been reported [68]. In addition, the interleukin 13 (IL-13) peptide modified nanoparticles enhanced cellular uptake in U87 cells that was observed within 1 h [69].

### 3.7. Cs-SeNPs Passed through the Blood–Brain Barrier

Glioma occurs in the brain, where only a small numbers of drugs can penetrate through the blood–brain barrier for treatment. Previous studies have shown that the presence of a positive charge on the surface of the nanoparticle will support cell entry [50]. Therefore, 0.2% Cs-SeNPs attached to endothelial cells will probably pass across the blood–brain barrier via adsorptive-mediated transcytosis, resulting in a therapeutic effect [69,70]. To confirm the capability of 0.2% Cs-SeNPs to transport across the blood–brain barrier, an *in vitro* BBB model was established to investigate the permeability of SeNPs and 0.2% Cs-SeNPs. Coumarin-6-labeled SeNPs and 0.2% Cs-SeNPs were prepared and added into the luminal compartment of a transwell plate. Then, the media at 0, 0.5, 1, 4, 6, and 24 h post-incubation of the lower compartment were collected and analyzed to identify the coumarin-6 signal using a fluorescent microplate reader. The signal intensities at 0, 0.5, 1, 4, 6, and 24 h post-incubation of coumarin-6-labeled SeNPs were −0.65 ± 0.62, 0.21 ± 0.63, 0.78 ± 0.53, 0.56 ± 0.22, 0.24 ± 0.66, and −0.77 ± 1.81, respectively. Meanwhile, the signal intensities of coumarin-6-labeled 0.2% Cs-SeNPs were −0.17 ± 0.16, 0.52 ± 0.97, 0.553 ± 0.33, 0.76 ± 0.69, 1.79 ± 0.41, and 15.84 ± 2.51, respectively (Figure 8). The results imply that 0.2% Cs-SeNPs more effectively penetrate the blood–brain barrier than SeNPs do, especially at 24 h post-incubation. As a result, the chitosan-coated SeNPs are a beneficial alternative therapeutic agent for improving glioma treatment.

## 4. Conclusions

In this study, SeNPs were fabricated to obtain a homogeneous and narrow-size distribution nanoparticle. SeNPs were modified by coating them with chitosan to enhance their stability and induce a positive charge. Cs-SeNPs revealed a higher anticancer activity than SeNPs when observed in glioma cells. The 0.2% Cs-SeNPs were selected to investigate the growth inhibition activity of glioma 3D-spheroid cells in combination with 5-FU. The 0.2% Cs-SeNPs exhibited a promising antitumor activity in terms of reducing tumor spheroid growth and increasing drug sensitivity to 5-FU. Furthermore, gelatin zymography also confirmed that 0.2% Cs-SeNPs can inhibit cell migration and cell invasion of glioma cells via reducing MMP-2/9 activities. This phenomenon might be due to increased cellular uptake of 0.2% Cs-SeNPs in U87 cells caused by the presence of chitosan in SeNPs, as confirmed by CLSM analysis. In addition, the coumarin-6-associated chitosan-coated SeNPs could cross the blood–brain barrier and enhance the therapeutic efficiency of SeNPs. The modification of SeNPs by coating them with chitosan demonstrated the benefits on physical properties, including enhanced SeNP stability and biocompatibility. The size changes due to the chitosan coating might be another factor that supports the cell entry of the particles. Moreover, we determined the antitumor activity against glioma, one of the most aggressive cancers. The antitumor activity against glioma needs to be confirmed *in vivo* to ensure the efficacy of Cs-SeNPs as a promising alternative nanomedicine.

## Figures and Tables

**Figure 1 nanomaterials-12-02606-f001:**
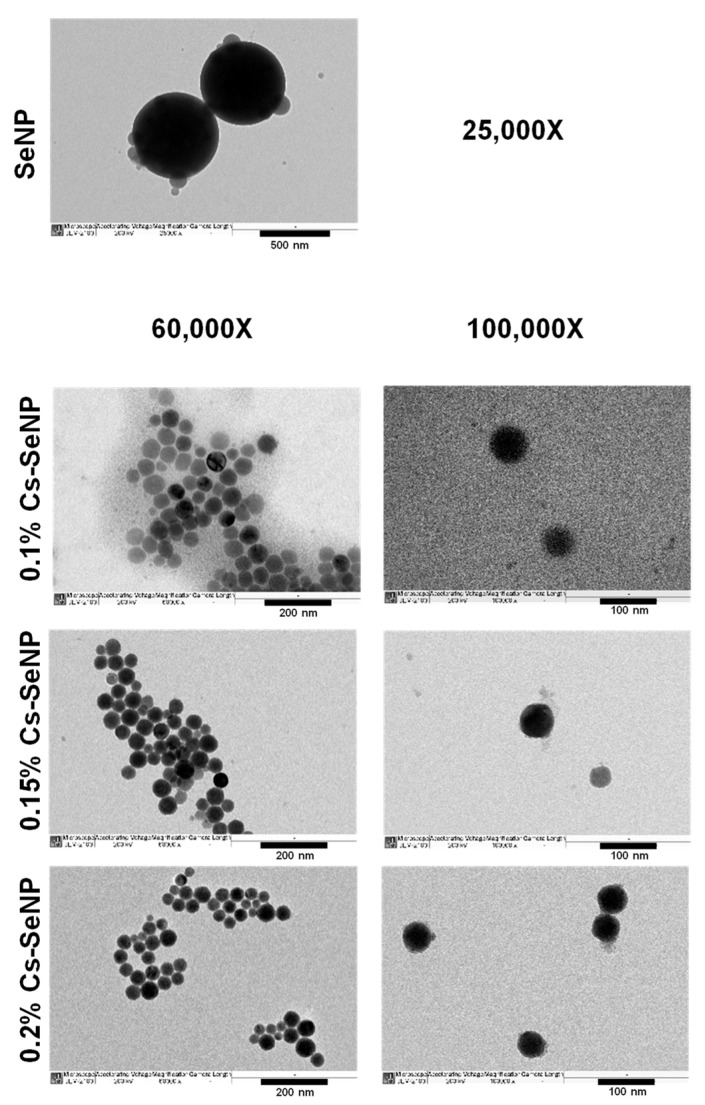
Morphology of selenium nanoparticles (SeNPs) and chitosan-coated SeNPs (Cs-SeNPs) analyzed by transmission electron microscope (TEM). Morphologies of the formulated Cs-SeNPs and SeNPs were observed under TEM with different magnifications (25,000× *g*, 60,000× *g*, and 100,000× *g*).

**Figure 2 nanomaterials-12-02606-f002:**
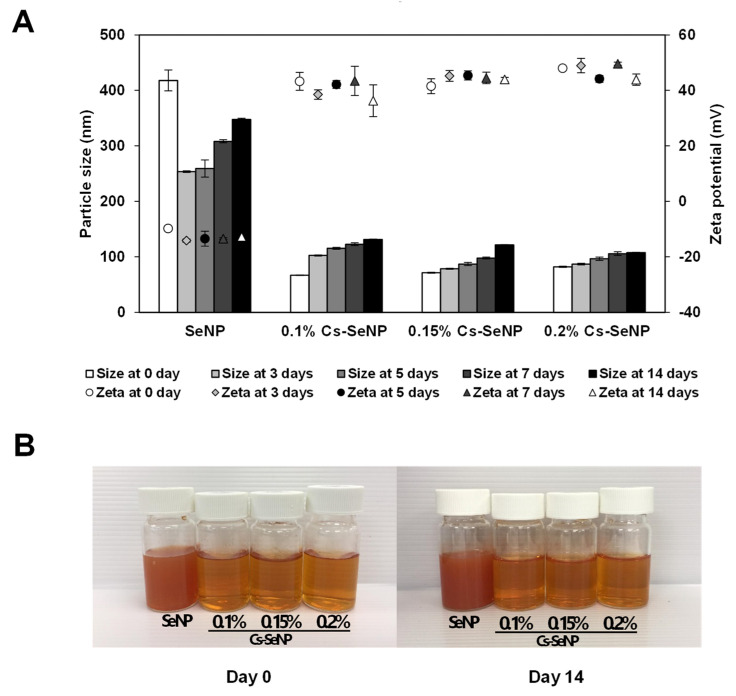
Stability of the chitosan-coated SeNPs and SeNPs after storing for 14 days at 4 °C. Stabilities of Cs-SeNPs and SeNPs were monitored by hydrodynamic diameter and zeta potential value. (**A**) The graph presents hydrodynamic diameter and zeta potential values determined using dynamic light scattering (DLS). (**B**) Gross appearance of the Cs-SeNPs and SeNPs at days 0 and 14.

**Figure 3 nanomaterials-12-02606-f003:**
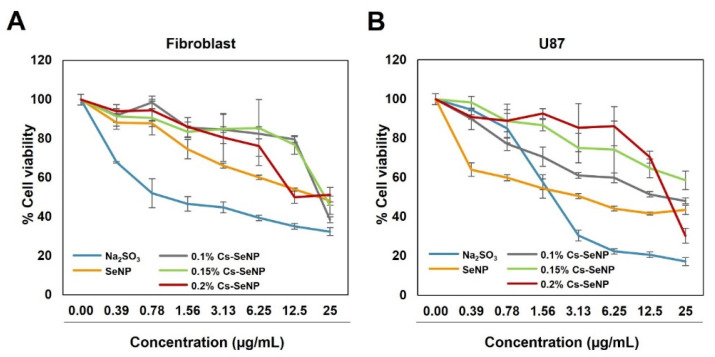
Cytotoxicity of free sodium selenite (Na_2_SeO_3_), SeNPs, and Cs-SeNPs against the U87 glioma cell line. (**A**) Cell viability of human normal fibroblasts and (**B**) U87 glioma cells exposed to the chitosan-coated SeNPs, SeNPs, and free sodium selenite for 24 h. Cell viability was analyzed using the MTT assay. Data are shown as representative data of two independent experiments, and error bars represent SEM.

**Figure 4 nanomaterials-12-02606-f004:**
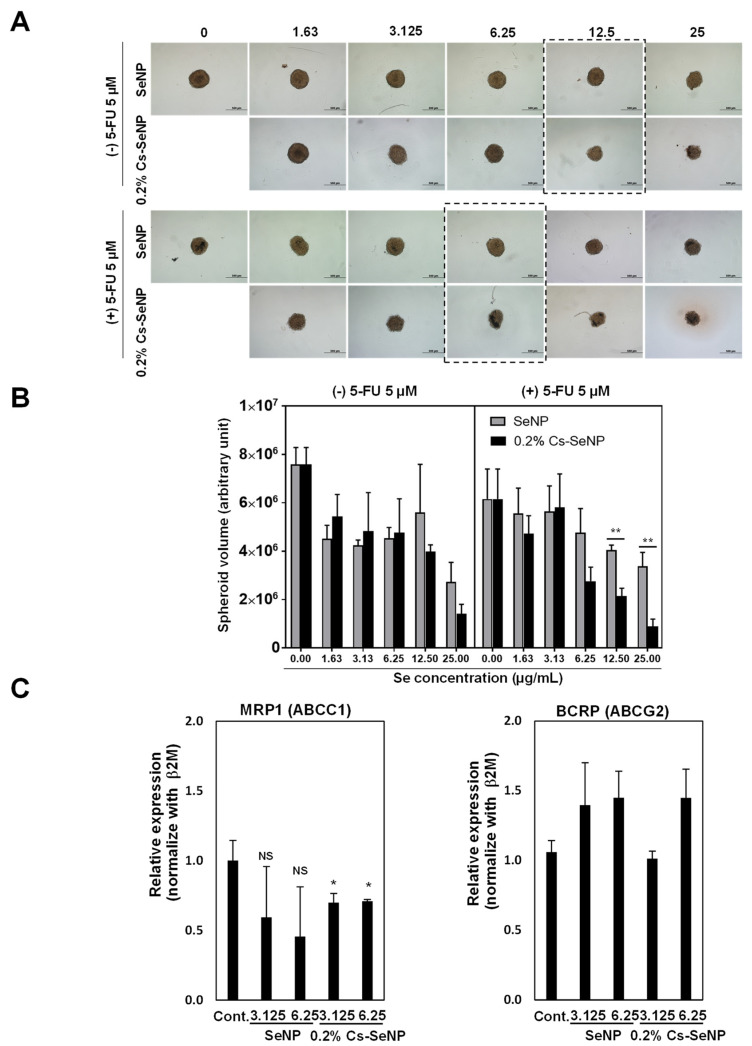
Effect of Cs-SeNPs on tumor spheroid growth of the glioma cell line with and without combining with 5-FU. (**A**) U87 tumor spheroid at 2 days old was exposed to Cs-SeNPs or SeNPs at various concentrations (0–25 µg/mL) for 72 h without 5-FU (upper panel) and with 5 µM of 5-FU (lower panel). Tumor spheroid morphology was observed under an inverted microscope at 10× magnification. Experiments were performed in triplicate. Dot boxes indicate the lowest concentration of nanoparticles that showed the obvious difference in tumor spheroid growth. (**B**) The quantitative analysis of tumor spheroid volume was calculated following the formular spheroid volume (V = 0.5 * Length * Width ^2^). The data are means ± SEM from the representative experiments. ** *p* < 0.01. (**C**) The ABC transporters related to drug resistance in glioma cells, namely MRP1 and BRCP, were determined in U87 cells treated with Cs-SeNPs or SeNPs using quantitative real-time PCR. Relative mRNA expression was calculated (untreated control cells = 1). The data are means ± SEM from the representative experiments. * *p* < 0.05., NS: no significant difference.

**Figure 5 nanomaterials-12-02606-f005:**
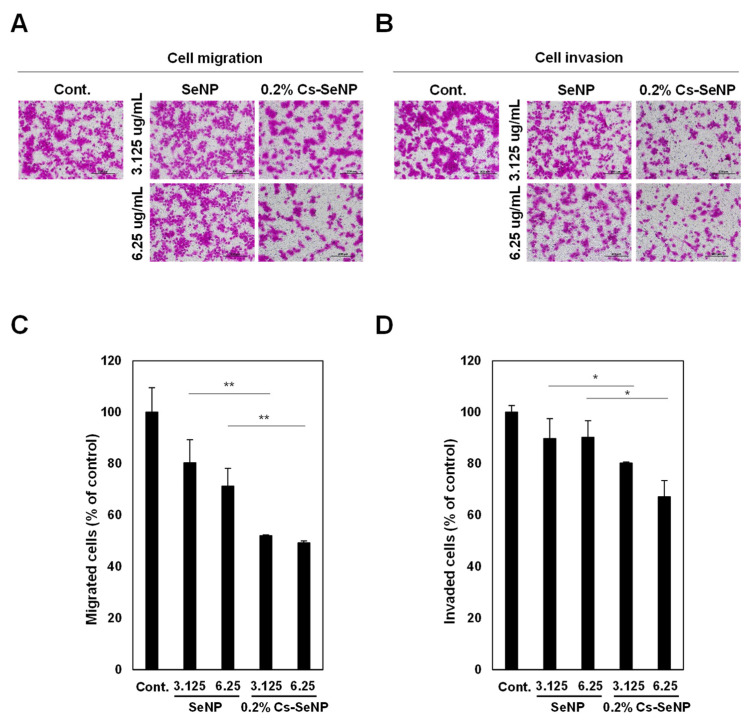
Effect of Cs-SeNPs and SeNPs on cell migration and cell invasion of U87 cells. Cells were treated with Cs-SeNPs or SeNPs at concentrations of 3.125 or 6.25 µg/mL for 24 h. (**A**) Migration and (**B**) invasion abilities of Cs-SeNPs-treated cells and SeNPs-treated cells were determined by Boyden chamber assay. (**C**) Percentages of migrated cells and (**D**) percentages of invaded cells were presented (untreated control cells = 100%). The data are means ± SEM from the representative experiments. * *p* < 0.05., ** *p* < 0.01.

**Figure 6 nanomaterials-12-02606-f006:**
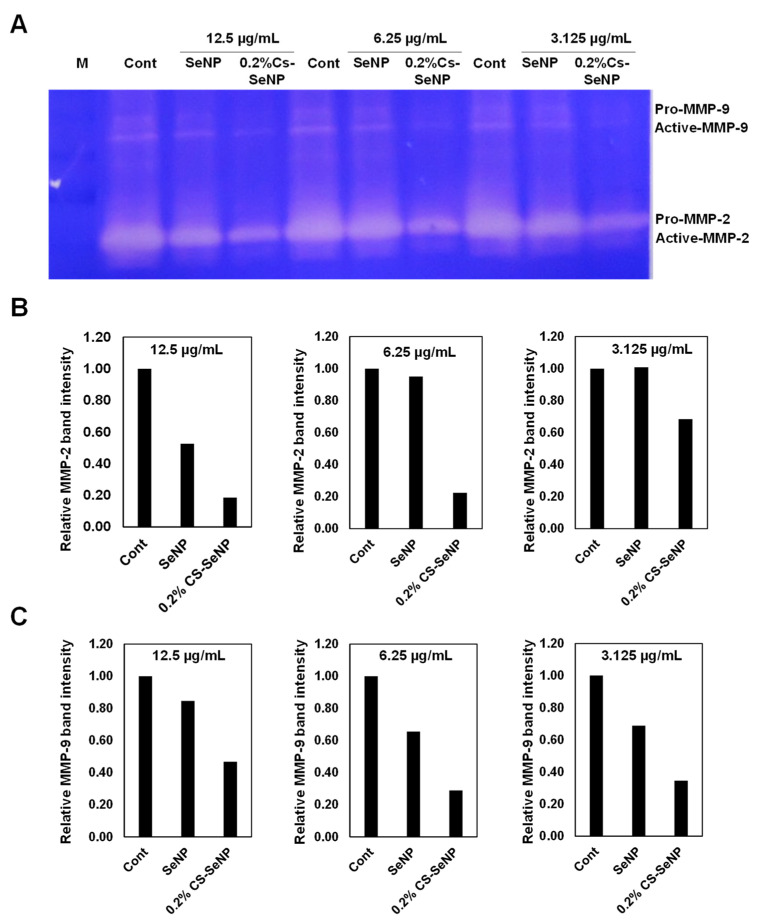
Effect of Cs-SeNPs on activity of MMP-2/9 in glioma cells. U87 cells were treated with Cs-SeNPs or SeNPs at concentrations of 3.125, 6.25, and 12.5 µg/mL for 24 h, and then the media were replaced with serum-free media and further incubated for 24 h. The conditioned medium was harvested and subjected to gelatin zymography assay. (**A**) A representative experiment of gelatin zymography assay, which revealed bands of MMP-2/9. Two experiments were performed, and (**B**) is a representative graph of quantitative analysis of MMP-2 and (**C**) MMP-9 band intensities from one experiment.

**Figure 7 nanomaterials-12-02606-f007:**
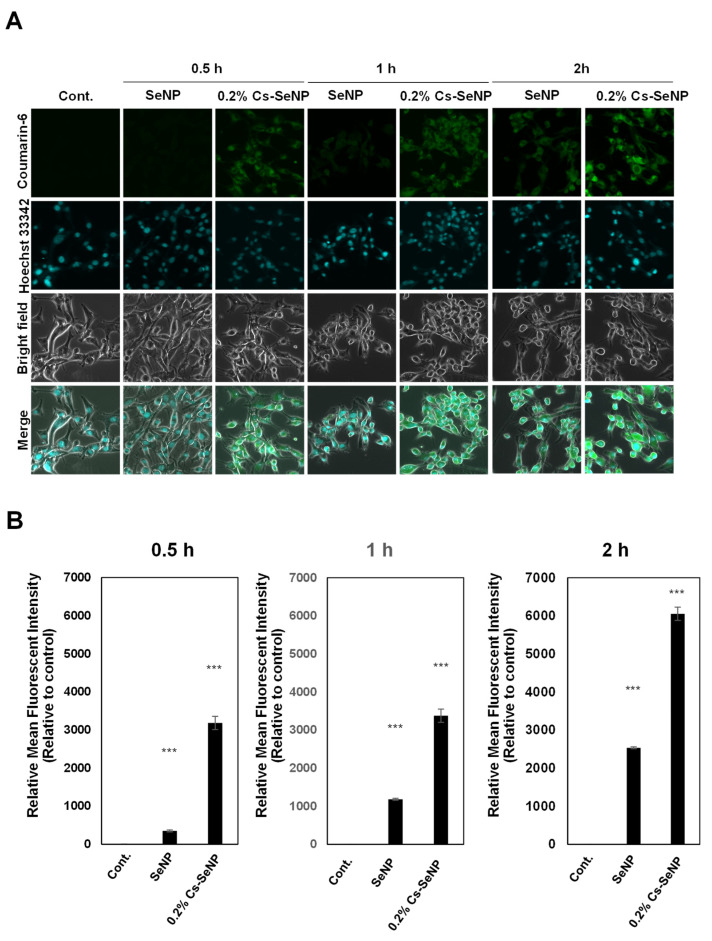
Cellular uptake of Cs-SeNPs and SeNPs by U87 cells. Cells were treated with coumarin-6-labeled Cs-SeNPs or coumarin-6-labeled SeNPs at a concentration of 6.25 µg/mL for 0, 0.5, 1, and 2 h. Cells were then fixed, and the signal of coumarin-6 was observed using CLSM. (**A**) Cellular uptake images were analyzed by CLSM. (**B**) Quantitative analysis of coumarin-6 signal in cells was performed using ImageJ software. Relative mean fluorescent intensity is shown in the bar graph (untreated control cells = 1). The data are means ± SEM from the representative experiments. *** *p* < 0.001.

**Figure 8 nanomaterials-12-02606-f008:**
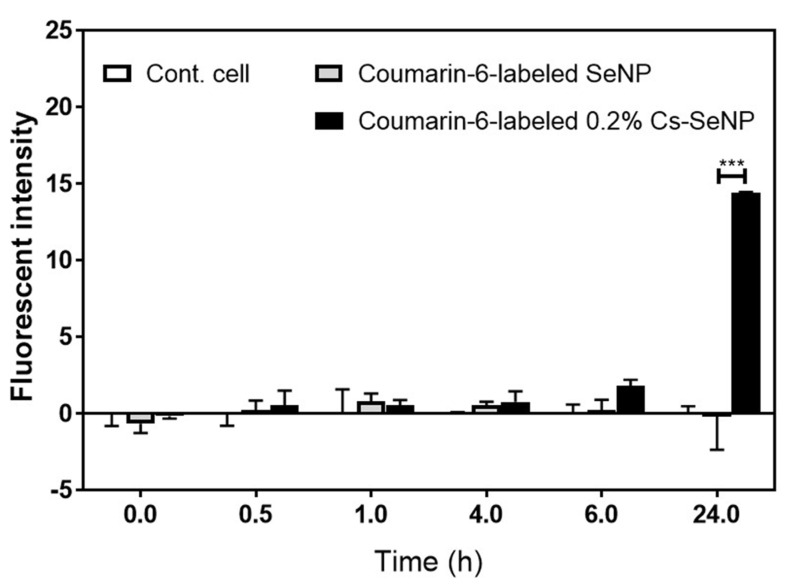
Ability of Cs-SeNPs to cross the blood–brain barrier. The bEnd.3 cells were seeded into the luminal side of the insert (pore size 0.4 µM). The lower chamber was filled with 700 µL of DMEM complete medium. Until the trans-endothelial electrical resistance (TEER) value was reached at 50 Ω·cm^2^, and coumarin-6-labeled Cs-SeNPs or SeNPs at Se concentration of 6.25 µg/mL were added into the luminal side of the insert for 0–24 h. At the indicated time point, the medium in the lower chamber (200 µL) was removed, and immediately the same volume of medium was added for further culture. The signal of coumarin-6 presented in the media was measured using a fluorescent microplate reader. The data are means ± SEM from the representative experiments. *** *p* < 0.001.

## Data Availability

The data are available on reasonable request from the corresponding author.

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
