# Peer review of "Inhibiting Metastasis and Improving Chemosensitivity via Chitosan-Coated Selenium Nanoparticles for Brain Cancer Therapy"

_nanomaterials, 2022, doi:10.3390/nano12152606_

Round 1

Reviewer 1 Report

1.     The Info about chitosan used in this study was missing.  

2.     Is there any better way to evaluate the viability of tumor spheroid?

3.     Could authors double-check the time of cellular uptake of U87? Can it happen in 1-2 hours?

4.     A few qualitative or semi-quantitative methods were used in this in-vitro study, such as measure the tumor spheroid, MMP2/MMP9 activity, which make the conclusion sounds too much optimistic. All the work was done in-vitro and I would suggest the authors to make appropriate conclusion based on the very limited work.

Author Response

Reviewer#1

  1. The Info about chitosan used in this study was missing.

Author: According to reviewer’s comment, the information of chitosan was added into section 2.1 materials, line 92.

  1. Is there any better way to evaluate the viability of tumor spheroid?

Author: There are some other approaches to determine cell viability of tumor spheroid for example ATP measurement. However, the method that we employed in this experiment is the way to investigate tumor spheroid growth and calculate tumor sphere volume. Based on this comment, we added in Fig. 4B to indicate the quantitative data obtained from spheroid assay.

  1. Could authors double-check the time of cellular uptake of U87? Can it happen in 1-2 hours?

Author: -Since the particle harboring the cationic charge from chitosan to facilitate the cell entry, so it is possible that chitosan-based nanoparticle can enter into the cells within hours. There are previously studies including  

- Preferential drug delivery to tumor cells than normal cells using a tunable niosome–chitosan double package nanodelivery system: a novel in vitro model.

This study indicated that the chitosan-based nanoparticle can entry into the glioma cell U373 at post incubation 20 min to 1 hour.

- Ligand modified nanoparticles increases cell uptake, alters endocytosis and elevates glioma distribution and internalization

This study showed that chitosan-based nanoparticle was able to enter U87 cell within 1 hour.

              Regarding to this comment, we referred to the previous study by added in line 437-440.

  1. A few qualitative or semi-quantitative methods were used in this in-vitro study, such as measure the tumor spheroid, MMP2/MMP9 activity, which make the conclusion sounds too much optimistic. All the work was done in-vitro and I would suggest the authors to make appropriate conclusion based on the very limited work.

Author: We modified the conclusion to be “The antitumor activity against glioma needs to be confirmed in vivo to ensure the efficacy of Cs-SeNP as a promising alternative nanomedicine.” Line 495-498.

Reviewer 2 Report

Please see attached file for comments.

Author Response

Reviewer#2

This manuscript presents the results of generating chitosan-stabilized selenium nanoparticles and

of several in vitro experiments using the nanoparticles with a human glioblastoma cell line.

Standard procedures were used to generate and characterize the nanoparticles, both with and

without chitosan. Several differences in biological effects concerning cytotoxicity, mRNA

expression of genes responsible for some types of chemotherapy resistance, and invasion were

observed in the GBM cells exposed to selenium nanoparticles relative to cells exposed to

chitosan-stabilized selenium nanoparticles. The chitosan-stabilized nanoparticles showed more

desirable effects for anti-cancer treatment relative to the non-stabilized nanoparticles in the

studies presented with the GBM cells. The results were as expected based on previous studies

using similar nanoparticles in other types of cancer cells. Overall, there is clear discussion of the

data. The following items are recommended to be addressed.

Comments for authors to address:

-Were spheroids treated with 5-FU alone? Having a comparison of cells treated with 5-FU

without the nanoparticles is important before it can be concluded that nanoparticles increased

sensitivity of the cells to 5-FU chemotherapy (lines 25, 81-82, 300, 313-316, and 446).

Author: We have tested spheroid without 5-FU at concentration of 5 µM and 10 µM compared to the control cells. The tumor spheroid volumes were 618441.54± 80305.8 and 680300.23± 128030.38 of 5 µM and 10 µM treated cells, respectively. Meanwhile, volume of control cell was 691902.33± 115148.5. These concentrations of 5-FU had no effect on tumor spheroid growth. In the manuscript this explanation was mentioned line 325-327.

-Have the authors determined if coumarin-6 “leaks” from the nanoparticles into solution? The

results in Sections 3.6 and 3.7 assume that coumarin-6 is still associated with the nanoparticles,

but no data is presented to indicate that the nanoparticles themselves are associated with the

fluorescence signal (or within the cells).

Author: With the present data, we have no experiment to determine the leakage of coumarin-6 dye. Suppose that if the leakage occurs, the signal intensity might be equally detected between SeNP and Cs-SeNP. However, according to the data obtained from confocal microscopy, the signal of coumarin-6 at the upper panel indicated the different level of coumarin-6 depending on cell entry.  As a result, coumarin-6 signal intensity from Cs-SeNP was higher than that of SeNP.

-In Section 3.2, the authors indicate that “0.2% Cs-SeNP showed the strongest effect on reducing

cell viability of U87 and the weakest effect on normal fibroblast cells” (lines 288-289). However,

the data presented in figure 3 seem to indicate lower cell viability of fibroblasts relative to U87

cells with those nanoparticles, except at the concentration of 25 uM. The authors should clarify

what is meant by their statement, in order to better match the data presented. (Typically, a

desirable chemotherapeutic agent and dosage is selected by the largest difference in survival

between normal cells and cancer cells, where normal cells remain viable and cancer cell viability

is significantly lower. Data in Figure 3 suggest the 0.1% Cs-SeNP would fit this criteria better

than 0.2% CS-SeNP.)

Author: We understand the reviewer point. However, we aim to select based on the criteria that 0.2% CS-SeNP provided the big different of toxicity between normal and cancer cells. Since at the highest concentration (25 ug/ml) 0.2% CS-SeNP showed lowest cell viability more than the other %CS-SeNP. Meanwhile, the lowest effect was found in normal fibroblast. The explanation was added in the manuscript Line 302-307.  

-Line 27, line 450, and lines 454-455: recommend changing the wording that SeNP could cross

the blood-brain barrier (BBB) because the presented studies did not directly determine if the

nanoparticles crossed the blood-brain barrier; an in vitro model of endothelial cells was used, but

that is not the same as the blood-brain barrier. And as stated above, it is not clear of the

nanoparticles, or simply the fluorescent dye, passed through the endothelial cell layer.

Author: According to reviewer’s suggestion, we changed to be coumarin-6 associated SeNP in Line 27 and revise wording to be line 493-498.

-Figure 1 images: the scale bars for the TEM images are very difficult to see, even when the

figure is enlarged; please present scale bars with more clarity.

Author: We modified according to the suggestion.

-Figure 2 image A: the scale bar for zeta potential appears to not match the data described in

section 3.1 (lines 236-238). Also, the legend does not indicate the symbols for data points of zeta potential measured on days 7 and 14. Please revise Figure 2A and the text so they are consistent.

Author: Fig. 2 A was revised, and We corrected as Line 248.

-In section 3.3 the authors discuss results presented in Figure 4; however, in the opinion of this

reviewer, the differences in size of the tumor spheroids are difficult to observe visually. Were the

sizes of the spheroids measured, so that quantitative data of spheroid growth inhibition could be

presented?

Author: Regarding to the reviewer ‘s comment, we quantitated volume of spheroid and added in Fig. 4B and added the data in text in section 3.3, line 333-342.

-Lines 193-194: it is recommended to change the wording that the concentration was non-toxic

to cells; cell survival less than 100% (even if still >80%) seems to indicate some level of

cytotoxicity, based on results in sections 3.2 and 3.3 (Figures 3 and 4).

Author: We edited according to the comment, to be “lower than IC20 concentration”.

-Please explain in the figure caption for Figure 4 what is meant by the dotted box around some of

the spheroid images in Figure 4 (boxes of spheroids with 12.5 uM NP with -5FU, and 6.25 uM

NP with +5FU).

Author: We added in Figure legend in line 375-378.

-Figure 8: please update the scale on the vertical axis to indicate the scale below 10. Also, the

data has negative fluorescence values, so this should be shown on the graph by adjusting the

scale to include the entire range of fluorescence values that were determined during the

experiment. The Cs-SeNP data points discussed in the text (lines 425-426) do not match the

graph; please ensure the data and graph are consistent.

Author: Fig. 8 was revised.

-The authors indicated that the differences in the effects on cells between selenium nanoparticles

and chitosan-stabilized selenium was explainable by the presence of chitosan. However, since

the different nanoparticles had significantly different sizes, it is possible that the effects were

size-dependent, rather than chitosan-dependent. Literature studies have indicated that

nanoparticles with different sizes, regardless of surface functionalization, induce different effects

in cells (such as PMID:18654486, PMID:16608261). It is recommended that the authors state

this as an alternative explanation during the discussion or conclusion.

Author: Thank you very much for this comment, we agree that the size changing by coating CS is also another factor that support the cell entry.  We added this explanation in conclusion section, line 493-494.

Additional minor comments and recommended changes:

-Line 62: recommend providing the full wording for Se nanoparticles at first mention of

abbreviation, which should be in parentheses (SeNP)

Author: We corrected as suggestion in line 62.  

-Section 2.2.2 (lines 100-112) did not specify formulation of SeNP without chitosan, this could

be added (also for 2.2.3)

Author: We separately mentioned SeNP synthesis, line 102-107.

-Line 227: “CS-SeNPs” should be changed to “Cs-SeNPs”

Author: We corrected, line 238.  

-Lines 258-260: the wording “charge of size” is not clear; was this intended to be “change of

size”? Please revise the wording.

Author: We corrected, line 271.

-Line 279: suggest revising the start of the sentence to the following: “The viability of U87 cells

treated with SeNP…”

Author: We revised according to the comment, line 293.  

-Line 301: suggest changing wording about 5-FU as the “most important chemotherapeutic

drug”, because several other chemotherapeutic drugs are also widely used to treat patients,

especially those with glioma.

Author: We revised according to the comment, line 319.    

-Section 3.4: it might be helpful to modify the wording to indicate that the percentage values of

migration and invasion are relative to the control cells.

Author: The wording was modified in line 389.

-Line 389: recommend replacing or deleting “genolytic”.

Author: The word was deleted, Line 424.

-Line 390: Please state in the legend if graphs of C from Figure 6 are from one experiment,

similar to what was stated for graphs in B from Figure 6.

Author: The figure legend was modified.

-Line 437: Please revise the following to clarify the wording: “The media was measured the

signal of coumarin-6 using fluorescent microplate reader.”

Author: We modified to be “The signal of coumarin-6 presented in the media was measured using fluorescent microplate reader.”, line 476-477.

-Lin 513: Please add the page number or article number to Reference 20

Author: We added.

Reviewer 3 Report

The paper titled is 'Inhibiting Metastasis and Improving Chemosensitivity via 2 Chitosan-Coated Selenium Nanoparticles for Brain Cancer 3 Therapy' is a very interesting read. 

Here are a few suggestions:

1. The effect of Cs-SeNP on tumor spheroid growth should be performed using a quantitative assay to ensure changes in cell viability. The size of the spheroid is not enough as the proliferating layer of cells around the spheroid maybe targeted by Cs-SeNP rather than the spheroid. Also, performing this in triplicates is not enough as it is not a good representation of the toxicity response.

2. The spheroid model needs to be validated to ensure that the gene expression changes for e.g., in terms of CYP expression are actually reflective of a 3D model.

 3. Please check key for figure 2A - seems to be incomplete.

4. Please check sentence structure throughout.

Author Response

Reviewer#3

  1. The effect of Cs-SeNP on tumor spheroid growth should be performed using a quantitative assay to ensure changes in cell viability. The size of the spheroid is not enough as the proliferating layer of cells around the spheroid maybe targeted by Cs-SeNP rather than the spheroid. Also, performing this in triplicates is not enough as it is not a good representation of the toxicity response.

Author: Based on this comment, we added in Fig. 4B and line 333-342 to indicate the quantitative data obtained from spheroid growth assay.

              As the triplicate experiment, we tested and found the dose dependent in responding to drug concentration and detected the toxicity response, so this data should be a representative data for the study.

  1. The spheroid model needs to be validated to ensure that the gene expression changes for e.g., in terms of CYP expression are actually reflective of a 3D model.

Author: Thank you for your suggestion. However, in this step, we focused on the effect of the formulated particle on drug resistant related genes at first. Your comment is benefit for further exploration in the deep details of drug metabolizing enzymes.

  1. Please check key for figure 2A - seems to be incomplete.

Author: Fig. 2 A was revised by complete the the symbols for data points of zeta potential measured on days 7 and 14. We corrected as Line 248 to make consistency in text and Fig. 2A. 

  1. Please check sentence structure throughout.

Author: English proofreading has been done.
